# Understanding Braess' Paradox in power grids

Benjamin Schäfer [1,2,3,4,11] ✉, Thiemo Pesch [5,11], Debsankha Manik[4,6,11], Julian Gollenstede[7], Guosong Lin[7], Hans-Peter Beck[7], Dirk Witthaut [8,9,11] & Marc Timme [4,6,10,11] ✉

The ongoing energy transition requires power grid extensions to connect renewable generators to consumers and to transfer power among distant areas. The process of grid extension requires a large investment of resources and is supposed to make grid operation more robust. Yet, counter-intuitively, increasing the capacity of existing lines or adding new lines may also reduce the overall system performance and even promote blackouts due to Braess' paradox. Braess' paradox was theoretically modeled but not yet proven in realistically scaled power grids. Here, we present an experimental setup demonstrating Braess' paradox in an AC power grid and show how it constrains ongoing large-scale grid extension projects. We present a topological theory that reveals the key mechanism and predicts Braessian grid extensions from the network structure. These results offer a theoretical method to understand and practical guidelines in support of preventing unsuitable infrastructures and the systemic planning of grid extensions.

Electrical power grids are among the largest and most fundamental technical systems of our time, foremost due to their broad range of enabling functions[1]: Almost all our infrastructures and activities, from communication to transport and from food supply to health care, crucially depend on a reliable supply of electric power. Due to the rapid increase of distributed renewable power generation[2], electrified vehicles[3], and sector coupling technologies[4,5], the topologies of power grid networks are strongly changing. In particular, extensions and reinforcements of power transmission grids are instrumental to the transition toward sustainable energy supply, requiring multi-billion investments[6–8]. In particular, future power networks will be increasingly relying on long-distance transport via high-voltage-directed-current (HVDC) lines, for instance, to connect offshore wind

generation with inland industrial sites or large urban areas. As a consequence, grid operation will change comprehensively, requiring thorough and robust planning of all measures and processes involved.

While a substantial increase in power transmission capacities seems inevitable to meet future demand[9,10], already individual measures may introduce novel threats and risks. A cornerstone manifestation of threats results from Braess' paradox, originally found in economics[11,12]. The counter-intuitive finding that network extensions or reinforcements may deteriorate a system's functionality was first described half a century ago by Dietrich Braess for traffic networks[11–14] and has since been termed Braess' Paradox. The paradox has been analyzed for a variety of different systems, including traffic systems[15], automatons[16,17], mesoscopic electron transport[18], tabletop electric or

[1]Institute for Automation and Applied Informatics, Karlsruhe Institute of Technology, 76344 Eggenstein-Leopoldshafen, Germany. [2]Faculty of Science and Technology, Norwegian University of Life Sciences, 1432 Ås, Norway. [3]School of Mathematical Sciences, Queen Mary University of London, London, United Kingdom. [4]Chair for Network Dynamics, Center for Advancing Electronics Dresden (cfaed) and Institute for Theoretical Physics, Technical University of Dresden, 01062 Dresden, Germany. [5]Forschungszentrum Jülich, Institute for Energy and Climate Research - Energy Systems Engineering (IEK-10), 52428 Jülich, Germany. [6]Network Dynamics, Max Planck Institute for Dynamics and Self-Organization (MPIDS), 37077 Göttingen, Germany. [7]Clausthal University of Technology Institute of Electric Power Technology (IEE), Clausthal-Zellerfeld, Germany. [8]Forschungszentrum Jülich, Institute for Energy and Climate Research - Systems Analysis and Technology Evaluation (IEK-STE), 52428 Jülich, Germany. [9]Institute for Theoretical Physics, University of Cologne, 50937 Köln, Germany. [10]Lakeside Labs, Lakeside B04b, 9020 Klagenfurt, Austria. [11]These authors contributed equally: Benjamin Schäfer, Thiemo Pesch, Debsankha Manik, Dirk Witthaut and Marc Timme. ✉e-mail: benjamin.schaefer@kit.edu; marc.timme@tu-dresden.de

mechanical systems[19], metabolic networks[20], sports[21], and microfluid flows[22]. Here, we use the term "Braess' paradox" in the following sense: Braess' paradox occurs when adding network capacity decreases network performance. The added network capacity can be a new line or an upgrade of an existing line. Meanwhile, the decreased performance manifests itself e.g., in terms of longer travel times (traffic), higher costs for participants (economics) or a higher load on specific parts of the network (flow) and thereby reduced stability. With Braess' paradox arising in these diverse contexts and forms, we have to ask: How and when do we observe this (generalized) Braess' paradox in power grid extensions? Can we develop tools to predict the occurrence of the paradox and understand it to guide operator decision?

Critically, Braess' paradox has been discussed and predicted in both DC and AC power systems[19,23–28]. However, most of these studies have been purely theoretical. The few experimental studies that investigated Braess' paradox used table-top settings and were constricted to small-scale, either DC or single-phase AC electric circuits. In particular, the paradox has not yet been identified or demonstrated in any large-scale real-world systems. Furthermore, a systematic understanding of when and how Braess' paradox emerges in power systems is still needed.

In this article, we demonstrate and analyze how Braess' paradox emerges in real-world electric power grids ranging from simple demonstrations to simulations based on real-world settings. We first experimentally demonstrate the paradox in the laboratory in three-phase AC networks of synchronous and virtual-synchronous machines, offering a direct analogy to those in full-scale power grids. Second, we develop an effective topological indicator to predict edges that exhibit Braessian flow changes. Third, we analyze the potential impact of Braess' paradox on the planned extension of continental power transmission grids. In particular, we identify Braessian edges in planned extension scenarios, highlighting the mutual impact of extensions planned by two European transmission system operators. For the full-scale German high-voltage grid, we employ a detailed model including unit commitment and power-flow studies, as also used by grid operators[7], to elucidate the core structural features that determine the uncovered Braessian responses to structural changes, going beyond the insights provided by common, purely numerical flow simulations. These results highlight Braess' Paradox as a prime example of a collective phenomenon required to be respected in temporally faithful and system-wide planning of grid extensions and for reliable grid operations.

## Results

Braess' paradox, as theoretically predicted for power systems[19,23–25,27,29] may emerge due to either upgrading existing lines or adding new lines, see Fig. 1: Upgrading (Fig. 1a) one line leads to a change $\Delta I$ of the current flowing across every edge. If this additional flow is aligned with the original flow on the line, the line loading increases and the upgrade potentially causes an overload and line shutdowns. Similar behavior is observed when adding a new line (Fig. 1b), where again, existing lines are subject to increasing current, see also Supplementary Note 4 for more details.

In the following, we address three joint questions about Braess' paradox for power grids that so far remained unanswered. First, is the paradox observable in laboratory-scale experiments with realistic parameters and in systems with synchronous and virtual machines as present in real grids? Second, how can we understand, effectively predict and potentially prevent the occurrence of Braess' paradox in power grids? Third, are current large-scale power grids and their already planned extensions susceptible to Braess' paradox?

### Experimental demonstration of Braess' paradox

We experimentally demonstrate the existence and fundamental properties of Braess' paradox in AC electric power grids in a laboratory-scale platform. The test grid is constructed out of two physical synchronous generators (Fig. 2a), as they are used in many power plants today[30], and two inverter-based nodes that mimic synchronous machines. Inverter-based solutions, such as virtual-synchronous machines (VISMA)[31–36] are expected to play an increasingly important role in the near future[37,38]. We remark that the synchronous generator is excited by an asynchronous motor, replacing, e.g., a steam-driven turbine that would drive the generator in conventional power plants. Braess' paradox was induced by changing the topology of the test grid using variable reactances and resistances, see Fig. 2b. In particular, we reduced the reactance $X_4$ from initially $X_4^{\max} = 0.25\,\Omega$ to $X_4^{\min} = 0.01\,\Omega$, increasing the susceptance $B_4$ and thereby upgrading line 4, by enabling additional flow through it, see Fig. 2c.

In contrast to naive intuition, upgrading a transmission line may not only upgrade but can equally downgrade the system's functionality. In the example laboratory setting, upgrading transmission line 4 increases the maximal load in the system: The grid extension decreases the load on the adjoining line 3 (green) but increases the load on transmission line 2 (orange), which is farther away. We consistently observe this response behavior in the laboratory experiment and theoretical estimates confirm this finding, a version of Braess' paradox, see Fig. 2. Since line 2 is not reinforced, the relative loading of this line as well as the dissipated electric power increases. A local increase of the load or dissipation can bring a transmission line to a critical state where it becomes prone to overheating or tripping, eventually threatening the operability of the entire network. We did not test for complete failure experimentally as this would destroy laboratory equipment. Additional experimental and computational results for

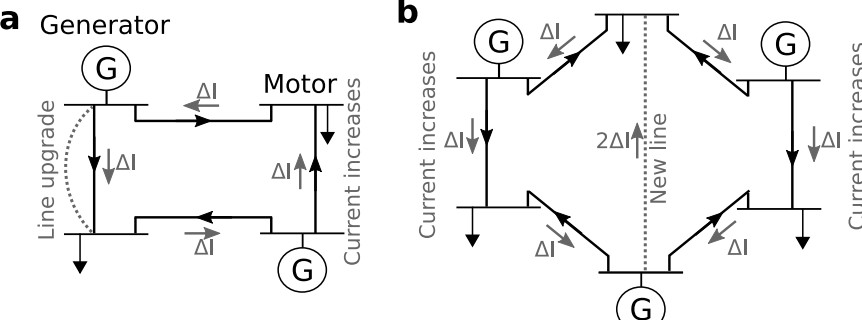

**Fig. 1 | Schematic of Braess' paradox emerging from upgrading or from adding transmission lines. a** Line upgrade in a four-node setup with two generators (G) and two loads, connected by a total of four lines. Original flows are indicated by black arrows and additional flows due to a line upgrade (dashed gray line) are marked by gray arrows and $\Delta I$. The current increases on one line, although that line is not upgraded. **b** Additional line in a six-node setup with three generators (G) and three loads causes increased current on two lines (right and left, labeled in gray). See Supplementary Note 4 for an additional discussion.

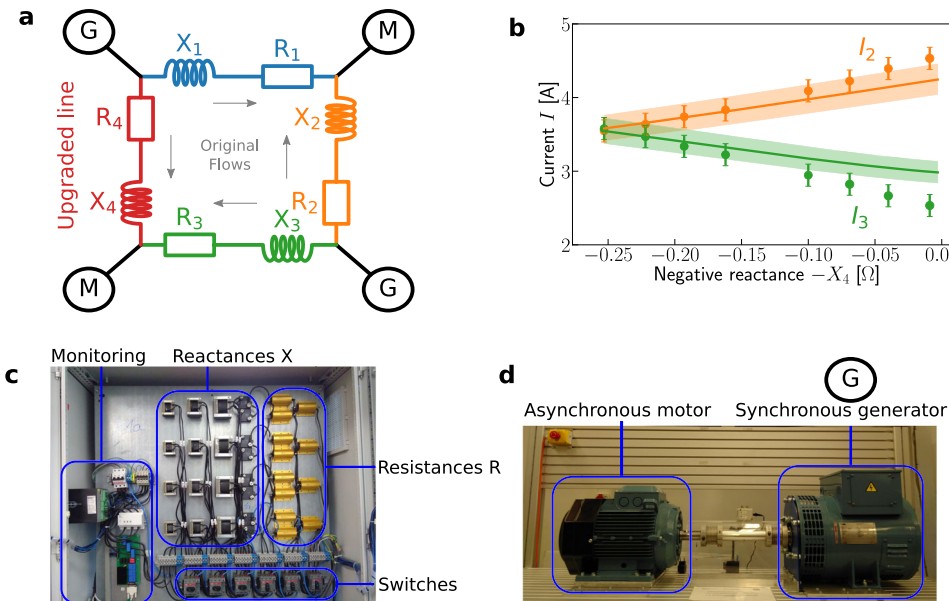

**Fig. 2 | Braess' paradox in laboratory-scale AC grids.** Here, Braess' paradox is observable as an increase of flow on the most highly loaded line due to upgrading a transmission line in the network. **a** Schematic of the experimental setup demonstrating Braess' paradox with two generators (G) and two motors (M). The reactance $X_4$ of line 4 is reduced, effectively upgrading the line. **b** The current amplitude $I_2$ and $I_3$ on lines 2 and 3 (as examples of lines where the additional flow is aligned and anti-aligned to the original flow) as a function of the negative reactance $-X_4$ of the upgraded line 4. While the current on line 3 decreases, line 2 carries an increasing load: Braess' paradox occurs. Dots with error bars and shaded regions

indicate average currents and their standard deviation based on measurement and estimation uncertainties. Solid lines indicate our theoretical predictions based on power-flow computations, see Methods. **c** Synchronous generators, driven by asynchronous motors, power the laboratory grid. Each node within the laboratory grid is either a synchronous machine or a virtual-synchronous machine (VISMA)[31]. **d** Line properties, i.e., resistances $R$ and reactances $X$ are freely tunable via switches in the laboratory grid. See Supplementary Note 1 for more details on the experimental setup and uncertainty estimation.

parallel lines for a six-node network and details on the synchronous machines are described in Supplementary Notes 1, 2, 4.

These experiments further highlight the basic mechanism causing Braess' paradox in AC power grids and underline the importance of the structure of a network's interactions and supply-demand distribution. Reinforcing one line increases its current and real-power flow, but also affects the flows on all other lines. With power injections at the nodes staying constant, the conservation of energy demands that the additional flow on the reinforced line is inducing a cycle flow, as demonstrated in Fig. 1. This additional cycle flow is necessarily aligned with the initial flow on the reinforced line. The flow changes on all other lines depending on their direction relative to the induced cycle flow. We use this idea of alignment and anti-alignment to define a predictor for Braessian edges in the next section.

We remark that our experimental four-node system is among the simplest settings to demonstrate Braess' paradox, and hence the paradoxical result might seem intuitive to some experts: Our theoretical predictions correctly capture the increase in current, as observed in the experiment. This alignment of theory and experiment is necessary but not sufficient to understand Braess' paradox. In larger networks, more complex interactions will be at work and it will not be obvious whether an additional line is improving or deteriorating overall performance. In particular, moving beyond single-loop networks, multiple cycle flows can be induced such that the current changes will no longer be symmetrical. Computing all currents for all scenarios for any network extension will reveal those scenarios that reduce overall performance but will not explain them. Before considering the complex network extensions below, it is thus critical to have validated the basic principles at work in an experimental setup. Hence, the four-node system discussed above serves as a comprehensible starting point, where experiment and theory readily align. In the following two sections, we build upon this starting point toward a general classification algorithm and more realistic application scenarios.

## Flow alignment predicts Braess' paradox

How can we predict whether a line upgrade would cause Braess' paradox, i.e., increase the load on other lines in larger networks? Although Braess' paradox has been reported in a plethora of contexts[19,23,27,29], a topological understanding of which *edges* are likely to induce Braess' paradox has proven elusive. Here, we demonstrate the first step towards bridging this gap.

We introduce a topological criterion that predicts how the modification of one edge $(s, t)$ affects the flow on another edge $(u, v)$. This approach is applicable to any pair of edges, but we focus on the edge $(u, v)$ with the highest load for the time being. This reflects the fact that lines with the highest initial load are most vulnerable to overloads. We then categorize an edge $(s, t)$ as Braessian if increasing its capacity induces an increase of load at the maximally loaded edge.

Let an edge $(u, v)$ exhibit the maximal load in a network and let the flow $F_{uv}$ be oriented from $u$ to $v$. If the capacity $B_{st}$ of another edge $(s, t)$, where the flow $F_{st}$ is oriented from $s$ to $t$, is increased infinitesimally, the change in the flow $F_{uv}$ is given by the edge-to-edge susceptibility[39] $\frac{dF_{uv}}{dB_{st}}$. The edge $(s, t)$ is Braessian if and only if the flow *change* across the maximally loaded edge $(u, v)$ is in the same direction (aligned) as the original flow across $(u, v)$, i.e., from $u$ to $v$. Figuring out if this is the case is analogous to finding the direction of currents in a resistor network, see Supplementary Note 3 for details. In this analog of resistor networks and here specifically, we propose the following heuristic, inspired by the electric lemma, popularized by Shapiro[40,41], connecting the currents in resistor networks with the numbers of spanning trees with certain properties in a network: Identify the shortest path from $t$ to $s$ that includes the edge $(u, v)$. If $u$ comes before $v$ in this path, the infinitesimal flow change $dF_{uv}$ is directed from $u$ to $v$, i.e., the flow change is aligned with the path.

We visualize the idea of alignment in Fig. 3. If the initial flow is anti-aligned, i.e. opposite to the new cycle flow, the resulting flow decreases. (Fig. 3a). Vice versa, if the flow is aligned, i.e., was originally in the same

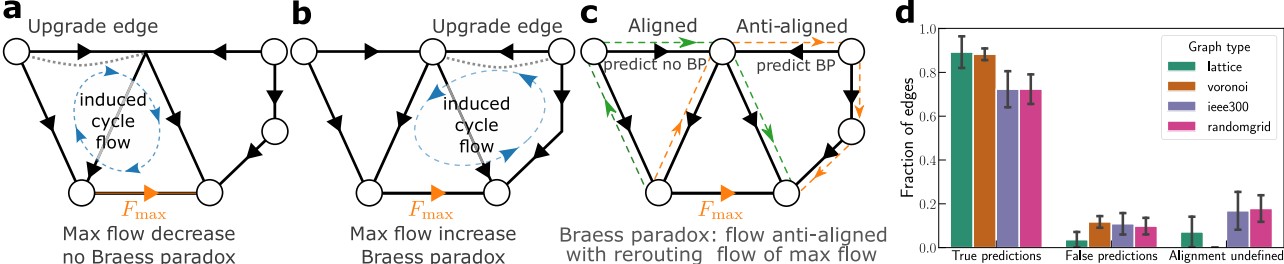

**Fig. 3 | Predicting Braessian edges through topological features. a** Upgrading edge (2, 3) induces a cycle flow that is anti-aligned with the flow on the most highly loaded line, thus reducing its load. Hence, Braess' paradox does not occur. **b** Upgrading edge (3, 4) induces a cycle flow that does align with the flow on the most highly loaded line, thereby increasing its load. Hence, Braess' paradox occurs. **c** In any given network, we systematically search for the shortest rerouting path across the maximum flow that includes the edge of interest. If the flow across that edge is aligned with the rerouting path[47], we predict the edge to be Braessian, see refs. [60], [61] and Supplementary Note 3 for details. **d** The predictor is successfully applied in four network topologies: two-dimensional square lattices, Voronoi tesselations, the IEEE 300 bus test case, and random power grid models generated using ref. [43], using homogeneous, i.e., unweighted lines. We generate 200 generator and consumer distributions for each case. This analysis has very little (about 3−11%) false predictions and about 72−89% of Braessian line extensions are correctly identified. The remaining links have undefined alignment such that the predictor is not applicable for these. Edges with susceptibility smaller than $10^{-4}$ are excluded from this analysis because upgrading them has too little impact on the maximum flow. See Supplements for details on the implementation of the predictor.

direction as the newly added cycle flow, the flow increases (Fig. 3b), causing Braess' paradox. If the shortest cycle is not unique and instead there are different shortest cycles yielding opposite alignments, we say the alignment is undefined. We test this heuristic using extensive numerical simulations with the linearized power-flow (DC) approximation. Please see also Supplementary Note 3 for details about the algorithmic implementation and the evaluation statistics. To obtain comprehensive statistics, we consider the topology of an established test grid, a regular lattice structure, an ensemble of random planar networks generated by constructing the Voronoi tessellation of randomly placed nodes[42] as well as random power grid networks generated using the model developed in ref. [43], using homogeneous, i.e., unweighted lines. Furthermore, we enhance the statistics by sampling generation and demand randomly multiple times instead of using only one snapshot. As Fig. 3c illustrated, the proposed heuristic performs very well for all considered topologies. For the regular square lattices, we observe a failure rate of ~3%, which gets slightly higher for the non-regular topologies, ~10%. We correctly classify ~88% of Braessian additions for square lattices and Voronoi tesselations, as well as ~72% for the other two topologies. For the remaining lines, alignment is undefined. False predictions can be attributed to cases where several paths of similar length contribute to the rerouting process such that the shortest path is not always the dominant one. Furthermore, paths may interfere, leading to surprising collective effects[41].

We note that our topological approach is complementary to established numerical methods, improving our general understanding of the impacts and benefits of grid extensions. Power system analysis frequently uses line outage distribution factors (LODFs), which predict the change of power flows after a line outage from a matrix algebraic computation[44]. These distribution factors can be generalized to arbitrary changes of the grid parameters, including transmission line updates[45,46]. But still, their computation relies on large-scale matrix algebra, thus restricting human insights. The electric lemma rigorously links the topological and the algebraic approach[40,41], but is hard to use in practice. Our approach approximates this lemma and provides an easily applicable predictor as well as an intuitive mechanistic picture of flow rerouting.

## Planned grid extension causing Braess' paradox

Braess' paradox may be the rule rather than the exception, as previous theory, together with the mechanism underlying our heuristic predictor, indicate. In particular, Braessian edges are by no means limited to laboratory-scale experimental setups and, as we demonstrate in the following, indeed emerge in large-scale AC power grids—even for currently ongoing extension projects. We analyze the German high-voltage power grid, for which a detailed, governmentally approved extension plan exists: the Netzentwicklungsplan (NEP)[7]. To quantify the impact of grid extensions, we employ a high-quality grid operation and unit commitment model using scenario data for 2020 (see Methods for details). Figure 4 illustrates the emergence of Braessian edges as a consequence of planned grid extensions. The examples are illustrated for a peak wind scenario in autumn: the hour with the highest wind power injection, which causes massive stress to the grid.

In particular, implementing two individual extensions induces Braess' paradox in the German power grid. Both extensions are currently under construction in the grid of the transmission system operator (TSO) Amprion, one AC extension and one construction of a new HVDC link. Both measures relieve the grid in the Northwest of the Amprion grid (see Fig. 4b for the base case without extension) but increase loads in the Northeast where the grid of two TSOs, Amprion and Tennet, are interconnected: This effect constitutes a classic example of Braess' paradox, which so far had not been identified in nation-wide grid extensions.

First, when implementing the AC extension projects AMP010 together with AMP011, both involving 380 kV lines around the city of Gütersloh, flows increase in the entire corridor Landesbergen–Ruhrgebiet. But not all lines in the corridor are extended and may thus suffer Braess' paradox. Notably, the remaining lines mostly belong to the TSO Tennet, not to Amprion, and no extensions are planned even at later stages, see Fig. 4c.

Second, the most expensive and momentous projects of the NEP are several new high-voltage DC (HVDC) lines connecting regions with high wind power generation in Northern Germany to the centers of the load. One of these projects, referred to as "corridor A", links the city of Emden at the Northern Sea with the Ruhr area (Fig. 4d). The installation of these lines leads to immense overloading of the 220 kV AC line Emden-Conneforde, representing another classical example of Braess' paradox: The AC-DC converter stations in Emden attract immense real-power flows from the connecting AC lines, which cannot cope with the additional flow. In fact, the NEP foresees an extension of the line Emden-Conneforde to 380 kV. Braess' paradox will be avoided if this measure is implemented first, compare, e.g., refs. [47], [48]. But if it is delayed or fails, the entire HVDC link may become useless or even dangerous for grid operability.

Whereas extensive grid simulations that are thoroughly coordinated between two or more TSOs may, in principle, detect Braess' paradox, we are unaware that Braess' paradox has been reported to occur or considered in this or any other national or international grid extension plan. Our findings thus highlight a particular threat emerging from the collective, system-wide interactions of large-scale extensions of high-voltage grids and point towards particular care

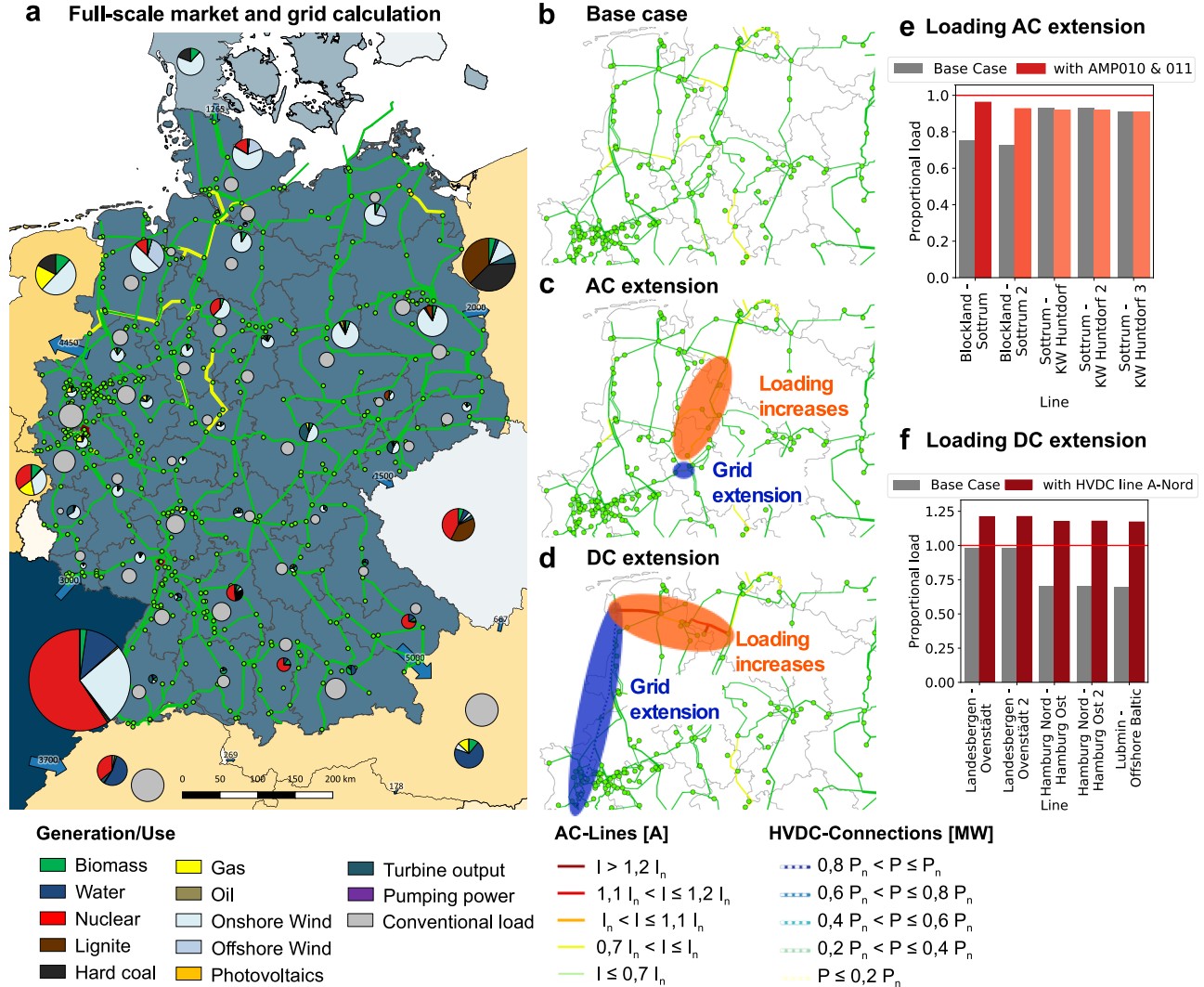

**Fig. 4 | Expansion plans in real power grids cause non-local overloads of the grid. a** A simulation of the German power grid is performed using a full-scale market where the color code shows the current relative to the thermal limit current: Yellow indicates high loads, while orange or red indicates an overload, see Methods for details on the simulation model. **b** Zoom on the North-western part of the German power grid in its base load case. **c** Including an AC expansion (blue oval) causes higher loadings (orange oval). The color code of the lines denotes the increase in load. Some lines are now close to their overload condition. **d** Including a long-range DC line (blue oval), again causes some lines to be close to their overload condition (orange oval). **e, f** We compare the proportional loading (actual loading divided by max load) of the most highly loaded lines before and after enhancing existing lines in both the AC (**e**) and the DC (**f**) extension scenario. The horizontal red lines indicate the transition to the overloaded state. $I_n$ and $P_n$ give the maximal current or power as designed for normal operation, i.e., any current $I > I_n$ or power $P > P_n$ signals an overload. Maps were created using the Quantum GIS Project and the Mapping Toolbox in MATLAB.

required in the choice and coordinated planning of the location and temporal order of such extensions.

The overload scenarios following a grid enhancement occur regularly and are again explained by induced cycle flows. In analogy to the arguments presented in Fig. 3, an extension in the German grid induces a cycle flow that, if aligned with the initial flow on the line, leads to an increased flow and potentially an overload, e.g., on the lines Blockland–Sottrum and Landesbergen–Ovenstädt for the AC and DC extensions respectively. To address the question of how frequently such flows may increase in real grids, we determine the $(N+1)$-criterion of a network: For a given line $e$, we compare its original current $I_e^{(0)}$ with the current $I_e^{(+a)}$, which is the current on line $e$ when doubling the capacity $B$ of line $a$. We consider this systematically for all lines $e$ in terms of their relative current change (Fig. 5a) as well as for one specific line, visualizing its absolute current (Fig. 5b). Enhancing a single line somewhere in the grid often leaves the current constant or induces only small changes (note the pronounced peak in the histograms at 0 relative current or at the initial current $I_e^{(0)}$). However, more severe

effects are also observed, where the (relative) current changes substantially. Interestingly, these changes are almost evenly split between improving or worsening the grid performance, thus causing many edges to be Braessian in the sense of increasing load. Importantly, we also observe cases, where the current of our unaffected line may increase beyond the security threshold $I_e^{th}$, making the network not $(N+1)$-secure (Fig. 5b).

## Discussion

In conclusion, we have demonstrated that Braess' paradox plays a pivotal role in real-power grids in several contexts. Specifically, our results on Braess' paradox advance the field in several directions: Beyond the original predictions for traffic flow[11,12] and a range of works on power grids and electrical circuits both theoretically[23–25,27,29,49] and in small-scale table-top experiments[19,29]. We demonstrated that Braess' paradox also robustly emerges in real-world electrical supply networks. In particular, we have shown that Braess' paradox emerges in laboratory grids involving synchronous machines and virtual-synchronous

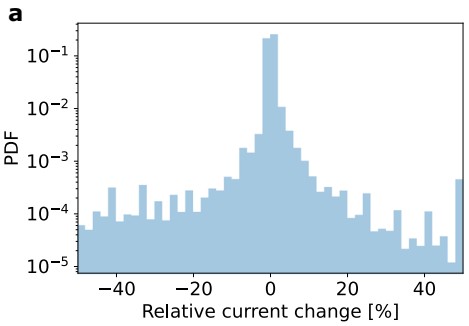

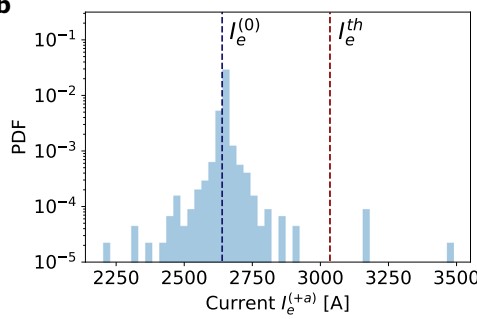

**Fig. 5 | ($N+1$)-extensions may induce overloads. a** We systematically consider all ($N+1$)-extensions and plot the relative change in current in a normalized histogram. **b** We display the absolute current $I_e^{(+a)}$ on one line $e$ when each line $a$ in the network is enhanced one by one in a normalized histogram. While most current changes are small (**a**), the new current $I_e^{(+a)}$ might surpass the current threshold $I_e^{th}$ of the line (**b**). Data derived from the market model as in Fig. 4 and considering all single line extensions provide the ($N+1$)-criterion.

machines settings as well as in planned and ongoing grid extension projects affecting national and potentially continental size grids.

An interesting question arising here is: How often do these Braessian effects already occur in real transmission grids and how do they affect the operational state? We note that the coupling of two busbars at Landesbergen transformer station, i.e., the addition of a new connection to the grid, played a crucial role during the 2006 European power outage[50]. With the heuristic predictor presented here, we offer topological insights and intuition into how network changes may strengthen network performance and avoid unintentional damage. This could be of particular importance when decisions have to be made in a short time, and no extensive simulations are feasible.

The precise extent to which extension projects induce problems via Braess' paradox is very difficult to gauge since not all extension plans are available to us. However, given the huge costs and long duration of grid extension projects, our finding strongly suggests to put an even larger emphasis on proper robust planning of extension projects, taking delays explicitly into account, including alternative or supportive extensions, and considering in detail the resulting collective system response at each state of the extension, as otherwise costly control measures will become necessary to keep the synchronous state[49,51,52].

In addition to establishing the emergence of Braess' paradox in relevant real grids, our results also offer a first tool for identifying and predicting Braess' paradox in flow networks. Whereas our predictor is heuristic in nature and by construction cannot be error-free, such a simple predictor using flow alignment is readily applicable across a range of flow networks, for instance, for microfluid flows[22] or traffic congestion[53]. We stress that the predictor allows identification and understanding of Braess' paradox from topological intuition in terms of cycle flows. It may thereby guide which additional connections to consider and which ones to avoid.

The observation of Braess' paradox both in a laboratory distribution and in a large-scale transmission system holds several important lessons for the effective planning of networked systems. First, systemic integrated planning is necessary, taking into account collective network phenomena such as the emergence of Braessian edges, in particular along corridors of grid extensions. The instance of Braess' paradox shown in Fig. 4b is rooted in the incomplete reinforcement of one corridor by only one of two TSOs. Second, temporarily integrated planning is necessary. The delay of one planned measure can render another measure useless or even dangerous (cf. Fig. 4c) even if the final stage of expansion has been suitably designed. As delays commonly occur when realizing large-scale infrastructure measures, planning for them is essential, also from the perspective of robust grid operation. Third, the ($N+1$)-criterion should be considered systematically when assessing the robustness of transmission grids.

The topological predictor presented here may serve as a simple and ready-to-use starting point to quickly assess these issues.

Our results also point to intriguing future research that could further explore potential benefits of the inverse of Braess' paradox, i.e. how to improve the state of the grid by removing lines to reducing loads, e.g., during cascading failures[54,55]. An intentional shutdown of lines could thereby decrease the current on other lines and protect the function of the entire system. In this context, the structural similarity between our predictor and line outage distribution factors (LODFs)[45,46] could be further exploited to identify lines that need to be disconnected to maintain a stable state. First results on inverse Braess' paradox stabilizing islanded grids have already been observed[56]. Moreover, the predictor could be developed further by including additional graph theoretical concepts and more complex computations. Furthermore, when deriving our predictor, we assumed infinitesimal line changes. A systematic study extending this to larger line modifications or line additions should be considered. Similarly, we chose to limit the scope of this present article to unweighted networks for the sake of clarity and brevity of the presentation. That said, the notion of flow alignment is readily extendable to weighted networks by computing shortest paths based on edge weights.

## Methods
### Power-flow computations
To compare the observed flows in the experiment with theoretical predictions but also when using the full market model, we compute flows by solving complex power-flow equations. In particular, we use the textbook form, see e.g., refs. 30, 44, 57, 58:

$$P_i = \sum_{j=1}^{N} E_i E_j \left[ G_{ij} \cos\left(\theta_i - \theta_j\right) + B_{ij} \sin\left(\theta_i - \theta_j\right) \right], \quad (1)$$

$$Q_i = \sum_{j=1}^{N} E_i E_j \left[ G_{ij} \sin\left(\theta_i - \theta_j\right) - B_{ij} \cos\left(\theta_i - \theta_j\right) \right], \quad (2)$$

where $P_i$ is the real power, $Q_i$ the reactive power, $\theta_i$ the voltage phase angle, and $E_i$ the voltage amplitude at each node $i \in \{1,...,N\}$. The line parameters are given by $G_{ij}$ and $B_{ij}$ as the conductances and susceptances respectively. When using $[E] = V$, $[P] = [Q] = W$ and $[G] = [B] = \frac{1}{\Omega} = \frac{W}{V^2}$, both sides of the equation use power in units of Watts.

In our experiment, we determined the line parameters as resistances $R_{lk}$ and reactances $X_{lk}$. Then, we compute the entries of the

nodal admittance $Y_{lk}$ for non-diagonal elements ($l \neq k$) as[57]

$$Y_{lk} = \frac{-1}{R_{lk} + \jmath \cdot X_{lk}}, \qquad (3)$$

with the imaginary unit $\jmath$. Finally, $G_{lk}$ and $B_{lk}$ have to be computed as

$$G_{lk} = \Re(Y_{lk}), \qquad (4)$$

$$B_{lk} = \Im(Y_{lk}), \qquad (5)$$

where $\Im(x)$ and $\Re(x)$ are the real and imaginary part of $x$, respectively.

### Simulation setup

Let us briefly review the power-flow model used to analyze the effects of Braess' paradox in a realistic power grid. As can be seen in Fig. 4, we focus on the German transmission grid, but also include trans-border power flows with neighboring countries. Electricity generation and demand are derived from a European electricity market model. While we provide an overview here, further details on the modeling are provided in ref. 48.

The European electricity market model includes the power dispatch planning on multiple time scales: Starting with a 1-year plan for the revisions of power plants, a day-ahead optimization is carried out to find the market-oriented power generation and usage of flexibility options within Europe on an hourly scale. In this (mixed integer programming) optimization, we explicitly consider technical constraints, such as minimum and maximum power, power gradients of individual plants, start-up processes, minimum running times as well as power withhold for control reserve. Special attention is paid to efficient modeling and solution of this optimization to include a large number of individual power plants, storage facilities, and flexibility options existing in the European power system.

The transmission grid model is a representation of the German transmission system at the extra high and high-voltage levels (110–380 kV), including equivalent circuits to model the lower voltage levels and surrounding grids. Data for the grid was made available by the German Federal Network Agency (Bundesnetzagentur). Crucially, data is not only available for the present state of the grid but also for several grid extension scenarios, including the addition of either HVDC or AC lines. The results of the simulation include not only power flows on all lines but also losses, all nodal voltages and currents, as well as the relative loading of all elements. Based on the domestic power dispatch and imports and exports, we can compute and visualize the loading of the grid with an hourly resolution for all possible extension scenarios.

### Data availability

A sample trajectory from the experiment for one set of parameters is uploaded at https://osf.io/s9fk2/, including the necessary evaluation software. The grid model can unfortunately not be disclosed as it involves data protected by a non-disclosure agreement. All further data that support the results presented in the figures of this study are available from the authors upon reasonable request.

### Code availability

Code to reproduce key results is available at https://osf.io/s9fk2/, as well as at ref. 59.

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

## Acknowledgements

We gratefully acknowledge support from the Federal Ministry of Education and Research (BMBF grants no. 03SF0472 and 03EK3055) (D.W., and M.T.), the Helmholtz Association (via the joint initiative "Energy System 2050—A Contribution of the Research Field Energy" and the grant no. VH-NG-1025 and no. VH-NG-1727) (D.W. and B.S.) and the German Science Foundation (DFG) by a grant toward the Cluster of Excellence "Center for Advancing Electronics Dresden" (cfaed) (M.T.). This project has received funding from the European Union's Horizon 2020 research and innovation program under the Marie Skłodowska-Curie grant agreement No 840825 (B.S.).

## Author contributions

B.S., T.P., D.M., J.G., G.L., H.-P.B., D.W., and M.T. conceived and designed the research. B.S., J. G., and G.L. conducted the experiments and analyzed the experimental data. T.P. performed simulations on realistic grid settings. D.M., D.W., and M.T. provided theoretical results on the predictor. B.S. and M.T. provided theoretical results on upgrading transmission lines in the laboratory grid. All authors contributed to discussing and interpreting the results and writing the manuscript. Overall, B.S., T.P., and D.M. contributed equally, as did D.W. and M.T.

## Funding

## Competing interests
The authors declare no competing interests.
