## [Peer Review File · Nature Communications]

REVIEWER COMMENTS

Reviewer #1 (Remarks to the Author):

The paper demonstrates a simulation method to show impact of Braess' Paradox in large power networks. The experiments give interesting results. In general, I liked the paper. I have few questions. I believe the answering this questions in appropriate places will make the paper more logical and coherent to read.

1. The predictor developed here is based on heuristic. Do the authors think there is a possibility of developing more robust method of predicting Braess' paradox in future?
2. What are the challenges of deriving a mathematical formula of existence of Braess' paradox in a large network and its associated damages/losses?
2. How can this paper help solve real world power grid extension problems? For any network extension, can we use this method directly and minimize the cost? Is the method plug and play type, meaning any point of time a new extension is planned on the existing network, can we check the effectiveness using your method?

Reviewer #2 (Remarks to the Author):

The paper is concerned with electrical power grids.

The authors elucidate the following situation: An additional link is added or a link is improved, and Kirchhoff's law implies that the flow on an existing edge is increased such that it is no longer admissible. A partial overload of the power network may be the result although the total loss of the energy in the augmented network may decrease.

The authors do not consider the public planning and security rules of the transmission system operators. This leads to an incorrect assessment of the current grid development plans and indication of supposed risks for system security.

The described effect differs from the special situation in traffic networks that is called Braess' paradox.

The losses on all paths for one origin-destination pair increase in that case.

There is a big difference between the two situations. In particular, the effect of Kirchhoff's law presented above is not paradoxical. The notation of Braessian edges is misleading in this case. The same holds for the title of the paper.

The referee recommends the rejection of the paper.

Reviewer #3 (Remarks to the Author):

See attached pdf.

Reviewer #4 (Remarks to the Author):

Beautiful paper. Few remarks:

1) caption of fig 4: "full-scale market where The color code" either break the sentences or uncapitalize "The"

2) fig 4: in AC extension is difficult to "read" where the blue oval is. Either zoom in or use filled ellipses in all the pictures (perhaps using transparency and/or putting ellipses in the background)

3) clarify what "ambiguous flow direction" means

4) very often Braess paradox is associated with the ADDING of a line, i.e. to a topological change. specify whether your predictors (i.e. both $dFuv/dBst$ and the cycle trick) work when adding a NON INFINITESIMAL new line. This could be also the subject of a new paper but must be clarified

5) as in point (4), specify in the discussion whether your predictors allow to correctly predict the lines to shut down to mitigate a blackout (again, since it is a topological, non-infinitesimal change, this could be an issue) or if it is an interesting direction to investigate in future work. However, to back up your intuition, you could cite that the inverse Braess paradox has already been observed in islanding simulations of the German power grid:

"We observe the Braess paradox in the inverse direction: removing lines can increase the number of working nodes on the islands, although in general it is the exception rather than the rule."

in

Mureddu, M. et al. "Islanding the power grid on the transmission level: less connections for more security" Sci Rep 6, 34797 (2016). <https://doi.org/10.1038/srep34797>

Response to the comments by the reviewers

We would like to thank all reviewers for their detailed feedback, interest and suggestions. All substantial changes to the manuscript and its Supplementary Information have been marked in blue and are also cited here for convenience.

Response to Reviewer #1

Reviewer Overview. The reviewer positively summarizes our paper, raising some questions to be addressed before publication.

The paper demonstrates a simulation method to show impact of Braess' Paradox in large power networks. The experiments give interesting results. In general, I liked the paper. I have few questions. I believe the answering this questions in appropriate places will make the paper more logical and coherent to read.

Author Reply: We thank the reviewer for their positive assessment of our work and reply to each question below.

Reviewer Comment 1. The reviewer wonders whether our heuristic predictor of Braess' paradox could be made more robust.

1. The predictor developed here is based on heuristic. Do the authors think there is a possibility of developing more robust method of predicting Braess' paradox in future?

Author Reply: This is a very good point raised by the reviewer. First, we would like to clarify that the main point of our heuristic predictor is that it allows us to gain insight from topological intuition. That means we can understand when and how Braess' paradox occurs via cycle flows and hope that this intuition and understanding prove useful in practice. Second, more robust methods are certainly possible within a follow-up by including more detailed graph theoretical tools and computations. We have clarified both points in the revised manuscript:

"We stress that the predictor allows identification and understanding of Braess' paradox from topological intuition in terms of cycle flows. It may thereby guide which additional connections to consider and which ones to avoid.

...

Moreover, the predictor could be developed further by including additional graph theoretical

concepts and more complex computations.”

Reviewer Comment 2. The reviewer inquires about the mathematical challenges arising when classifying Braessian lines.

2. *What are the challenges of deriving a mathematical formula of existence of Braess’ paradox in a large network and its associated damages/losses?*

Author Reply: This is a very relevant question. Actually there exists an exact criterion for whether an edge is Braessian or not. However, in order to actually use this criterion one needs to compute all spanning trees in a graph, which makes it computationally infeasible and therefore not relevant for larger networks in practise. We outline it below.

We have shown in Supplementary Note 3, Algorithm 1 that determining whether an edge is Braessian is equivalent to figuring out the direction of current across that edge due to a single dipole current source placed across the maximum flow F_{max} . Shapiro [1] showed that the direction of current across any edge (a, b) in a DC network due to a single source s and a single sink t can be computed as follows. Let the number of spanning trees that contain a path from s to t of the form s, \dots, a, b, \dots, t (i.e. traversing a before b) be $N_{a,b}$. And let $N_{b,a}$ be the number of spanning trees that contain a path from s to t of the form s, \dots, b, a, \dots, t . Then if $N_{a,b} > N_{b,a}$, the current is from a to b , and if $N_{a,b} < N_{b,a}$, the current is from b to a .

This exact criterion unfortunately does not lead to any *topological insight* into which edges in a network are Braessian, since whether $N_{a,b}$ is larger than $N_{b,a}$ or vice versa cannot be deduced easily from the network topology for nontrivial networks, nor can it be reduced to any well known topological measure that we are aware of. We have added a reference to the exact result now, see modified paragraph below:

”In this analog of resistor networks and here specifically, we propose the following heuristic, inspired by the electric lemma, popularized by Shapiro [1, 2], connecting the currents in resistor networks with the numbers of spanning trees with certain properties in a network: Identify the shortest path from t to s that includes the edge (u, v) . If u comes before v in this path, the infinitesimal flow change dF_{uv} is directed from u to v , i.e. the flow change is *aligned* with the path.”

Reviewer Comment 3. The reviewer asks how our method can be applied to real-world

power systems.

3. How can this paper help solve real world power grid extension problems? For any network extension, can we use this method directly and minimize the cost? Is the method plug and play type, meaning any point of time a new extension is planned on the existing network, can we check the effectiveness using your method?

Author Reply: Indeed, as pointed out by the reviewer, our method could be applied as stated in the revised manuscript:

”The topological predictor presented here may serve as a simple and ready-to-use starting point to quickly assess these issues.”

Connected to the first point above, we stress again that the main contribution of this (heuristic) predictor is its added understanding of the system, guiding which kinds of extensions to plan and which ones to avoid in the planning. This complements full scale simulations of the whole systems, which might yield more precise results but offer little in terms of explanations and understanding. We have included the following to stress this point in our discussion:

”With the heuristic predictor presented here, we offer topological insights and intuition into how network changes may strengthen network performance and avoid unintentional damage. This could be of particular importance when decisions have to be made in short time and no extensive simulations are feasible.”

Response to Reviewer #2

Reviewer Comment 1. The reviewer interprets Braess' paradox as closely related to the original traffic system. They point out that the interpretation of increased load on one line, given the enhancement of another line, as Braess' paradox in power systems is not immediately clear.

The paper is concerned with electrical power grids. The authors elucidate the following situation: An additional link is added or a link is improved, and Kirchhoff's law implies that the flow on an existing edge is increased such that it is no longer admissible. A partial overload of the power network may be the result although the total loss of the energy in the augmented network may decrease. The authors do not consider the public planning and security rules of the transmission system operators. This leads to an incorrect assessment of the current grid development plans and indication of supposed risks for system security.

The described effect differs from the special situation in traffic networks that is called Braess' paradox. The losses on all paths for one origin-destination pair increase in that case.

There is a big difference between the two situations. In particular, the effect of Kirchhoff's law presented above is not paradoxical. The notation of Braessian edges is misleading in this case. The same holds for the title of the paper.

The referee recommends the rejection of the paper.

Author Reply: We apologize for not being fully clear on how we understand "Braess' paradox" within our work. For us, as for a substantial part of the existing literature, Braess paradox is connected to the idea that the addition of a new link leads to a network with worse performance: Longer travel times in the case of traffic, higher prices in the case of economics or higher load and thereby lower stability in flow networks. We have specified this in the revised manuscript: "Here, we use the term "Braess' paradox" in the following sense: Braess' paradox occurs when adding network capacity decreases network performance. The added network capacity can be a new line or an upgrade of an existing line. Meanwhile, the decreased performance manifests itself e.g. in terms of longer travel times (traffic), higher costs for participants (economics) or a higher load on specific parts of the network (flow) and thereby reduced stability."

Hopefully, the reviewer agrees that in this sense, we indeed observe Braess' paradox in power grids as additional capacity (increasing the current a line can carry) or new lines do

decrease the performance in the sense of higher total loads and less secure operation. With this context specified, we also rephrased our key questions:

”With Braess’ paradox arising in these diverse contexts and forms, we have to ask: How and when do we observe this (generalized) Braess’ paradox in power grid extensions? Can we develop tools to predict the occurrence of the paradox and understand it to guide operator decision?”

Finally, we note that Dietrich Braess himself (after whom the paradox is named) lists several articles regarding Braess’ paradox on his webpage, see <https://homepage.ruhr-uni-bochum.de/dietrich.braess/#paradox>. This includes the article [3], which discusses Braess’ paradox in power grids in the same sense as our present article.

Response to Reviewer #3

Reviewer Comment Overview. The reviewer summarizes our main findings and suggests some modifications, in particular an increased emphasis on the predictor and the theoretical results.

The authors focus the well known Braess' paradox in the context of power grids, where it may arise due to grid extensions in the form of additional lines or line capacity. The crucial findings are: (1) an empirical evidence of the Braess' paradox in a lab setting (2) a mathematical heuristic to predict edges whose capacity increase could create the Braess' paradox and (3) a numerical study and detailed discussion of two line extensions for German power grid that caused such phenomenon. I personally find the empirical evidence mentioned in (1) limited and not very insightful (see also the related major comments below), to the point that I'm wondering whether is appropriate for it to have such a central position in the manuscript. The numerical study behind contribution (3) is interesting and shows the great relevance that this theory could have for power extension planning. However, it seem a bit "narrow" and it is too disconnected from the theory developed in (2). On the other hand, the idea behind (2) is intriguing, but I wished it was expanded much more and made more rigorous and general. I feel many technical details or the reasons behind them are omitted in this manuscript, which would instead greatly benefit from them. In the major comments/issues I suggest several other ways to improve the manuscript. I believe the work that the authors need to put into this paper to address all these issue is substantial, but probably still within the scope a major revision – this is my recommendation to the AE.

Author Reply: We thank the reviewer for their detailed assessment of our work and reply to each point below. As we cannot cover every technical detail here, we also refer the reviewer to [4], which we also cite now in the revised manuscript.

Major comments/issues

Reviewer Comment 1. The reviewer suggests to rephrase the title to clarify the focus of the study.

I find the phrasing of the title a bit misleading. Braess' paradox per se does not constrain power grid extension (we could build the "wrong" lines if we wanted to), but rather power grid extensions should be informed by and account for Braess' paradox risk? I do not have

a concrete suggestion for a better title, though.

Author Reply: We thank the reviewer for this comment. A main contribution of our paper is to point out that Braess' paradox occurs both in experimental and theoretical settings and we can understand it using efficient estimators. To reflect this, we have modified the title to

Understanding Braess' Paradox in power grids

Reviewer Comment 2. The reviewer asks for further explanation of the structure of the article, in particular on the role of the experimental results.

I wonder if the experimental demonstration of the Braess' paradox truly deserves the prime spot in the result section? Being a theory person myself, I am personally less interested in such an empirical evidence. If the authors consider this to be the key result, then I would say that the extent of this empirical evidence should be broadened, for instance by considering larger and more complex network topologies. I do find the current one extremely simple and symmetric.

Author Reply: We thank the reviewer for this comment. We selected this structure for the following reason (and made our best efforts to reflect this in the revised manuscript): Starting with a simple experimental system, we showcase the emergence of Braess' paradox, followed by theoretical understanding of the experimental observations and finally completed by detailed numerical simulations. The experimental demonstration also serves to justify that the theoretical and numerical results are not just artifacts of the model used. Finally, with this structure, we increase the degree of complexity of the system where Braess' paradox emerged continuously and keep the whole manuscript as open as possible for the interdisciplinary audience of Nature Communications. We have added several comments in the manuscript to highlight and justify this structure and cite two instances here:

"With Braess' paradox arising in these diverse contexts and forms, we have to ask: How and when do we observe this (generalized) Braess' paradox in power grid extensions? Can we develop tools to predict the occurrence of the paradox and understand it to guide operator decision?"

...

In this article, we demonstrate that and analyze how Braess' paradox emerges in real-world electric power grids ranging from simple demonstrations to simulations based on real-world

settings”

Reviewer Comment 3. The reviewer suggests to comment on Braess’ paradox in asymmetric settings.

Related to the experimental setting and the previous point, is there an instance of a network of Braess’ paradox where the increase of a current on one line is not identical to the decrease of current on another line? Maybe with less symmetric topologies this can happen? I would find such an empirical result much more compelling than the current one.

Author Reply: This is an excellent point. Indeed, in the experiment discussed, we observe symmetric changes in the current. This is to be expected from Kirchhoff’s law for any single loop cycle flow. Meanwhile, for more complex topologies, involving multiple cycles and thereby cycle flows, the current increase and decrease on lines will no longer be as symmetrical. While, we do not have enough machines available to demonstrate this in the experiments, we discuss this point in the paper more explicitly:

”In particular, moving beyond single loop networks, multiple cycle flows can be induced such that the current changes will no longer be symmetrical.”

Reviewer Comment 4. The reviewer asks us to comment on changes in the load distribution throughout the network, e.g. due to seasonal variations of demand and generation.

A more high-level “philosophical” question, which I still consider important for the motivation of the paper. The occurrence of Braess’ paradox heavily depends on the spatial distribution of generation and demand. A windy winter night scenario is radically different than a summer daily scenario with no wind, also in terms of demand. Different power line expansions can lead to Braess’ paradox depending of which of these scenarios we consider. In one case there could be no current increase, in the other one yes, making the Braess’ paradox very “situational”? Do the authors have a concrete recommendation when it comes to how to account for this huge variability in the power injection patterns? An insightful discussion on this matter could be a nice addition to the paper.

Author Reply: We thank the reviewer for raising this important point: The location of the most heavily loaded line and the direction of the flows before the grid extension of course depend on the power injections. Actually, the theory we developed is capable of predicting whether the flow across *any edge of interest* will increase due to strengthening one specific

edge or not. We have naturally focused on the most heavily loaded edge as the edge of interest. If the most heavily loaded line changes with time due to changes in power injections, we just need to apply our method to the new maximally loaded edge.

In the example in the paper we have focused on the most crucial case – the hour with the highest wind power injection which causes massive stress to the grid. We have now mentioned this in the article, as follows.

”Figure 4 illustrates the emergence of Braessian edges as a consequence of planned grid extensions. The examples are illustrated for a peak wind scenario in autumn: the hour with the highest wind power injection which causes massive stress to the grid”.

Reviewer Comment 5. The reviewer has a set of recommendations of how to further incorporate technical details.

I believe that many more technical details in pages 8 and 9 are needed. I discuss some of the issues below.

Author Reply: We thank the reviewer for this suggestion. More technical details have been added to the manuscript as requested, as well as the citation to a separate preprint[4] that focuses on the rigorous mathematical derivations of our predictor. We specify these in detail below.

Reviewer Comments 6&7. The reviewer asks for further explanations how deriving a heuristic predictor is beneficial.

Why do we need a heuristic to predict Braessian edges that is correct only 75-80% of the time? Is it so computationally expensive to run a DC power flow and check exactly which? The authors never discuss the computational complexity or running of the proposed heuristic, so appreciating its merits.

Author Reply: Thank you for drawing our attention to the fact to something that we did not emphasize in the article enough: The main contribution of this (heuristic) predictor is that it provides *topological understanding* of which edge extensions may have adverse effects. This complements full scale simulations of the whole systems, which might yield more precise results but offer little in terms of explanations and understanding. Such intuitive understanding may be valuable in various scenarios like (a) planning which kinds of extensions to realize and which ones to avoid, (b) while making operational decisions in

grid control stations based on live monitoring. For example, in 2006, a heuristic decision at Wehrendorf station triggered a cascade of failures leading to a major power outage. Indeed, the controllers decided to *add* a connection - they connected two busbars at Wehrendorf - which lead to an unexpected result, namely the overload of Landesbergen-Wehrendorf [5]. This can be counted as a prime example of Braess' paradox with far reaching consequences, and also for the non-trivial and sometimes counter-intuitive aspects of Braess' paradox. A full grid simulation would have obviously captured this effect, but the decision was reached by heuristic considerations within seconds to minutes. We have highlighted this further in the manuscript:

In the abstract we now state:

"We present a topological theory that reveals the key mechanism and predicts Braessian grid extensions from the network structure."

and in the discussion, we altered the following paragraph:

"An interesting question arising here is: How often do these Braessian effects already occur in real transmission grids and how do they affect the operational state? We note that the coupling of two busbars at Landesbergen transformer station, i.e. the addition of a new connections to the grid, played a crucial role during the 2006 European power outage [5]. With the heuristic predictor presented here, we offer topological insights and intuition into how network changes may strengthen network performance and avoid unintentional damage. This could be of particular importance when decisions have to be made in short time and no extensive simulations are feasible."

Related to the previous point, maybe the proposed heuristic allows to bypass the more expensive AC power flow calculations? If so, it's not clear and it's not been validated.

Author Reply: We hope that our response to the previous comment also answers this question.

Reviewer Comment 8. The reviewer suggests to clarify the definition of a Braessian edge. *I find the definition of Braessian edge confusing and possibly wrong. Why is the increase of load relevant only for the maximally loaded edge? What if the load on the original maximally loaded edge is unchanged but there is another new maximally loaded edge after capacity is increased? This is actually the case for the German grid study, cf. Fig.4 (see also bullet*

point below).

Author Reply: This is a valid point. If the capacity of an edge is changed by a non-infinitesimal amount (or an edge is added/removed completely), the maximal load in the network may change to another edge. Firstly, in order to avoid this complexity and to present the new concept in the first place, we have decided to focus on infinitesimal edge strengthening in the present article. Secondly, if the maximally loaded edge changes as a result of a grid modification, that does not pose any fundamental challenge to our method for predicting Braess paradox. We remark that, as we mentioned in our response to comment 4, our method based on flow alignments can be applied to *any edge*, not necessarily to the one that was maximally loaded prior to the strengthening. Indeed, if more than one “potentially vulnerable” line need to be monitored for potential increase in load, our approach is capable of doing so. To clarify our procedure in the revised manuscript, we have made a couple of changes including the following:

In the Results section (“Flow alignment predicts Braess’ Paradox”):

”How can we predict whether a line upgrade would cause Braess’ paradox, i.e. increase the load on other lines in larger networks? Although Braess’ paradox has been reported in a plethora of contexts [3, 6–8], a topological understanding of *which edges* are likely to induce Braess’ paradox has proven elusive. Here, we demonstrate the first step towards bridging this gap.

We introduce a topological criterion that predicts how the modification of one edge (s, t) affects the flow on another edge (u, v) . This approach is applicable to any pair of edges, but we focus on the edge (u, v) with the highest load for the time being. This reflects the fact that lines with the highest initial load are most vulnerable to overloads. We then categorize an edge (s, t) as *Braessian* if increasing its capacity induces an increase of load at the maximally loaded edge.”

In the discussion/outlook:

”Furthermore, when deriving our predictor, we assumed infinitesimal line changes. A systematic study extending this to larger line modifications or line additions should be considered. Similarly, we chose to limit the scope of this present article to unweighted networks for the sake of clarity and brevity of presentation. That said, the notion of flow alignment is readily extendable to weighted networks by computing shortest paths based on edge weights.”

Reviewer Comment 9. The reviewer wonders about the role of edge weights when predicting Braess' paradox.

9. Is the shortest path identification (which is the cornerstone of the proposed heuristic) considering edge weights or not? If unweighted, what is the reason behind this choice? If weighted, what are the weights and why? Physical lines properties (susceptances?), absolute flows, line loads, etc?

Author Reply: This is again a very valid point. We consciously chose to limit the scope of this present article to unweighted networks for multiple reasons. (a) Whether the flows across two edges are aligned or not – the central ingredient of our theory – is very easy to visually check for unweighted networks. (b) That said, the notion of flow alignment is very suitable for extending to the case of weighted networks. Since the notion is based on shortest paths (as we wrote in the article "...Identify the shortest path from t to s that includes the edge (u, v) . If u comes before v in this path, the infinitesimal flow change dF_{uv} is directed from u to v , i.e. the flow change is *aligned* with the path.") one needs to just compute the shortest paths taking edge weights into account. We have left this aspect out of the present article for sake of simplicity and brevity. We have added a short text to this effect in the Discussion section (as already stated above):

"Similarly, we chose to limit the scope of this present article to unweighted networks for the sake of clarity and brevity of presentation. That said, the notion of flow alignment is readily extendable to weighted networks by computing shortest paths based on edge weights."

Reviewer Comment 10. The reviewer suggests to point out cases where the heuristic predictor fails.

Did the authors have any insight in when their heuristic fails and why? Why are there undefined alignments? Are they "pathological cases" due to symmetries in the networks that would go away if weighted networks were considered? Or is a more obscure/fundamental issue?

Author Reply: Thank you for this interesting question. Indeed, we do have such insights. First, there are cases where the alignment is undefined. This can happen if the shortest cycle is not unique and if two of these shortest cycles indicate a different alignment. These cases are actually included in our statistical analysis and discussed in the text. This also

hints to cases where the predictor may fail: A failure can occur if there are two (or more) cycles which are of high importance for flow rerouting and of similar length. In this case it is possible that slightly longer cycles are more important. If the shorter, but less important path yields a different alignment, then the predictor fails. We have expanded this in the revised version:

”For the regular square lattices we observe a failure rate of $\sim 3\%$, which gets slightly higher for the non-regular topologies, $\sim 10\%$. We correctly classify $\sim 88\%$ of Braessian additions for square lattices and Voronoi tessalations, as well as $\sim 72\%$ for the other two topologies. For the remaining lines alignment is undefined. False predictions can be attributed to cases where several paths of similar length contribute to the rerouting process such that the shortest path is not always the dominant one. Furthermore, paths may interfere leading to surprising collective effects [2].”

Reviewer Comment 11. The reviewer urges us to make the choices of our test networks more transparent and reproducible.

Aiming to have the paper focused on power grid extensions, I find inappropriate the usage of Voronoi tessellation and 2D lattices as networks to test the quality of the predictor. I acknowledge the effort to choose planar graphs, but there are many established way in the literature to generate synthetic networks which topologically resemble real power networks. Furthermore, there is plenty of IEEE test networks available that the authors could use to perform a comprehensive statistical analysis without the need of random graphs? In any case, IEEE test networks or random graphs, many more details about this numerical study should be provided as to make them replicable. Furthermore, why are the networks taken to be unweighted? How are generation and demand sampled? Where are generators and load nodes placed on the non-IEEE networks? Are they fixed or their position is also re-sampled?

Author Reply: We thank the reviewer for these very valuable comments. Following their advice we have extended our paper as follows:

- We fully agree with the reviewer about the necessity of additional tests for typical power grid topologies. Hence, we added a new network topology to the mix: the random power grid networks generated using the model developed in [9].
- We also agree with the necessity of improving the transparency and reproducibility

of our study. We have made the full code available in [10].

We note that we have kept the result for the lattice and Voronoi in the paper. In the context of power engineering, these results are of minor relevance, but they additionally support our message. In the context of graph theory, they are valuable because they demonstrate the generality of the approach beyond specific classes of networks. Regarding why we chose to perform our analysis on unweighted networks, we would like to refer to our reply to Comment 9.

Reviewer Comment 12. The reviewer suggests to clarify the relation of our predictor and LODFs.

The claim at the end of page 9 that this work is complementary to LODFs since line are added instead of removed is very intriguing. But why didn't the authors develop an actual theory for line "addition" distribution factors instead of just proposing an heuristic? The theory of LODFs is very well understood and rich, under the DC approximation, so this LADF theory should be doable?

Author Reply: We thank the reviewer for this helpful question. We admit that our initial formulation was misleading at this point so we have completely rewritten this paragraph. Indeed, the topologic heuristic was never thought to replace the distribution factors for the analysis of grid extensions or failures. Instead, it shall complement numerical methods by providing a human-understandable mechanistic picture of the impacts of grid modifications. We stress this aspect in the new version of the manuscript:

"We note that our topological approach is complementary to established numerical methods, improving our general understanding of the impacts and benefits of grid extensions. Power system analysis frequently uses the line outage distribution factors (LODFs), which predict the change of power flows after a line outage from a matrix algebraic computation [11]. These distribution factors can be generalized to arbitrary changes of the grid parameters, including transmission line updates [12, 13]. But still their computation relies on large-scale matrix algebra, thus restricting human insights. The electric lemma rigorously links the topological and the algebraic approach [1, 2], but is hard to use in practice. Our approach approximates this lemma and provides an easily applicable predictor as well as an intuitive mechanistic picture of flow rerouting."

Reviewer Comment 13. The reviewer asks to clarify the role of the maximally loaded line

for the heuristic predictor.

Figure 4e reports the loading of the lines before and after a line extension. The lines which are reported to have a load increment are not the one that were maximally loaded. In fact, the line with supposedly the maximum load before the extension (chart only reports 6 lines) experiences a lighter load. This seems not to be aligned with the theory developed earlier in the paper, which focuses solely on the maximally loaded edge (admittedly so, cf. p8). Could your heuristic predict Braess' paradox for this line expansion, since it's not the maximally loaded line to experience it?

Author Reply: We again have to admit that our initial formulation was misleading at several points. Indeed, the heuristics works for any pairs of edges, one being modified and one experiencing a flow change. In the discussion and analysis of the topological predictor, we focused on the impact on the line with the highest initial loading for the sake of definiteness and simplicity. We have now revised and extended the introductory paragraphs of the section "Flow alignment predicts Braess' Paradox" to clarify this aspect (same text as cited in comment 8):

"How can we predict whether a line upgrade would cause Braess' paradox, i.e. increase the load on other lines in larger networks? Although Braess' paradox has been reported in a plethora of contexts [3, 6–8], a topological understanding of *which edges* are likely to induce Braess' paradox has proven elusive. Here, we demonstrate the first step towards bridging this gap.

We introduce a topological criterion that predicts how the modification of one edge (s, t) affects the flow on another edge (u, v) . This approach is applicable to any pair of edges, but we focus on the edge (u, v) with the highest load for the time being. This reflects the fact that lines with the highest initial load are most vulnerable to overloads. We then categorize an edge (s, t) as *Braessian* if increasing its capacity induces an increase of load at the maximally loaded edge. "

Reviewer Comment 14. The reviewer suggests to revise figure 5.

Figure 5 is not very informative, in my opinion. Which line e does this histogram refer to? How has this line been chosen? Do we a qualitatively similar histogram for all lines? If so, is there a more rigorous and quantitative way to report this finding rather than displaying an histogram for a single line?

Author Reply: Thank you for pointing this out. Indeed, the previous figure 5 was an illustration on one specific line and its absolute currents before and after a new line is added. We have revised figure 5 and kept this illustration as panel b, while a new panel a displays the systematic change in current for all lines and all line extensions. The revised figure and the revised text are given here in context:

”To address the question how frequently such flows may increase in real grids, we determine the $(N + 1)$ -*criterion* of a network: For a given line e , we compare its original current $I_e^{(0)}$ with the current $I_e^{(+a)}$, which is the current on line e when doubling the capacity B of line a . We consider this systematically for all lines e in terms of their relative current change (Fig. 1a) as well as for one specific line, visualizing its absolute current (Fig. 1b). Enhancing a single line somewhere in the grid often leaves the current constant or induces only small changes (note the pronounced peak in the histograms at 0 relative current or at the initial current $I_e^{(0)}$). However, more severe effects are also observed, where the (relative) current changes substantially. Interestingly, these changes are almost evenly split between improving or worsening the grid performance, thus causing many edges to be Braessian in the sense of increasing load. Importantly, we also observe cases, where the current of our unaffected line may increase beyond the security threshold I_e^{th} , making the network not $(N + 1)$ -secure (Fig. 1b).”

FIG. 1. $(N + 1)$ -extensions may induce overloads. **a**: We systematically consider all $(N + 1)$ -extensions and plot the relative change in current in a normalized histogram. **b**: We display the absolute current $I_e^{(+a)}$ on one line e when each line a in the network is enhanced one by one, in a normalized histogram. While most current changes are small (a), the new current $I_e^{(+a)}$ might surpass the current threshold I_e^{th} of the line (b). Data derived from the market model as in Fig. 4 and considering all single line extensions provide the $(N + 1)$ -criterion.

Reviewer Comment 15. The reviewer suggests to homogenize figure styles.

Networks are drawn in quite different ways across manuscript and supplemental material. I'm aware that the experimental setup requires a detailed network scheme, but I suggest the authors to at least uniformize all the other more abstract networks in style, layout and size as much as possible.

Author Reply: Thanks for the feedback. We have now homogenized the figures, e.g. by using non-serif fonts.

Reviewer Comment 16. The reviewer points out ways to improve Supplementary Note 1.

I suggest the reader to make Supplementary Note 1 more tidy, there is a lot of white space and figures of very different sizes in a layout/order which does not help the reader. Also, I would move this SN to be the last one?

Author Reply: As suggested, we tidied up Supplementary Note 1, reducing white space substantially and making the figures of more equal size.

There are probably too many decimal digits in Table II?

Author Reply: Indeed, the original number of decimal digits was a bit excessive. We have

reduced the number of digits displayed.

Minor comments:

- p7 r121 “acycle” → “a cycle” and there is an odd spacing between “Fig.” and “1”
- p12 “where The color” → “where the color”
- p13 r239 “we” → “We”
- p13 r250 “our finding strongly suggestS”
- p13 r250 “robust proper planning” → “proper robust planning”?
- p14 r271-272 I think the phrasing of this sentence could be improved (cf. the two adverbs “quickly” and “initially”)
- p15 r 292 Ylk is used before having been defined?
- In SM, the way the pseudocode is written in algorithm 1 can be greatly improved.

Author Reply: We thank the reviewer for pointing out these issues. We have fixed these, in particular, we defined Ylk before using it and revised the presentation of algorithm 1.

Response to Reviewer #4

Reviewer Comment 1. The reviewer recommends a revision of figure caption 4.

1) *caption of fig 4: "full-scale market where The color code" either break the sentences or uncapitalize "The"*

Author Reply: We apologize for the typo and have fixed it.

Reviewer Comment 2. The reviewer suggests to revise figure 4 for better readability.

2) *fig 4: in AC extension is difficult to "read" where the blue oval is. Either zoom in or use filled ellipses in all the pictures (perhaps using transparency and/or putting ellipses in the background)*

Author Reply: Following the reviewer's suggestion, we filled the ellipses, making them semi-transparent and made the labelling clearer (see next page)

Reviewer Comment 3. The reviewer asks us to clearly define the term "ambiguous flow direction".

3) *clarify what "ambiguous flow direction" means*

Author Reply: We apologize for not making this clear in our initial submission. Let us clarify (see also Supplementary Note 3 (especially lines 137-144)): We determine the maximum flow in the network and then check whether the additional cycle flow aligns with this original flow or not. Hence, the important distinction is between "aligned", "anti-aligned" and "alignment undefined". We use the category "alignment undefined" in the following cases: If there are two cycle flows in the network, one aligned and one anti aligned with the maximum flow, then our predictor cannot classify them as "aligned" or "anti-aligned". Such cases we refer to as "alignment undefined". Previously, we also used "ambiguous" but removed this word for clarity reasons. This has been clarified in the text (Supplementary Information):

"However, by sacrificing perfect accuracy in favour of simplicity, we can design an approximate topological predictor for Braessian edges inspired by the exact topological criterion [14]:

1. Let the maximum flow be across edge (u, v) , directed from u to v . Remove the edge (u, v) . If (u, v) is a bridge, then there is no Braess paradox in this network.

2. Our aim is to predict if another edge (s, t) is Braessian. The flow there is from s to t .
3. Remove the edge (s, t) . If it is a bridge, it is not Braessian.
4. Look at the shortest simple path $[u, \dots, s, t, \dots, v]$, i.e. the shortest simple path from u to v that contains the edge (s, t) and touches s before t . Let its length be l^+ .
5. Look at the shortest simple path $[u, \dots, t, s \dots, v]$. Let its length be l^- .
6. If $l^- < l^+$, predict that (s, t) is Braessian. If $l^- > l^+$, predict that (s, t) is not Braessian. If they are equal, then this predictor does not work (*Alignment undefined* in Figure 3 of the main text)."

FIG. 2. Expansion plans in real power grids cause non-local overloads of the grid.

Reviewer Comment 4. The reviewer suggests to discuss possible research questions when considering non-infinitesimal line changes in the predictor.

4) very often Braess paradox is associated with the ADDING of a line, i.e. to a topological change. specify whether your predictors (i.e. both dF_{uv}/dB_{st} and the cycle trick) work when adding a NON INFINITESIMAL new line. This could be also the subject of a new paper but must be clarified

Author Reply: Thank you for pointing this out. Indeed, so far, our predictor is based on the idea of infinitesimal changes and only complemented by the case studies of non-infinitesimal changes in the German grid. We have clarified the need for additional work in the outlook: "Furthermore, when deriving our predictor, we assumed infinitesimal line changes. A systematic study extending this to larger line modifications or line additions should be considered. Similarly, we chose to limit the scope of this present article to unweighted networks for the sake of clarity and brevity of presentation. That said, the notion of flow alignment is readily extendable to weighted networks by computing shortest paths based on edge weights."

Reviewer Comment 5. The reviewer points to additional literature that could be included in the discussion.

5) as in point (4), specify in the discussion whether your predictors allow to correctly predict the lines to shut down to mitigate a blackout (again, since it is a topological, non-infinitesimal change, this could be an issue) or if it is an interesting direction to investigate in future work. However, to back up your intuition, you could cite that the inverse Braess paradox has already been observed in islanding simulations of the German power grid:

"We observe the Braess paradox in the inverse direction: removing lines can increase the number of working nodes on the islands, although in general it is the exception rather than the rule." in Mureddu, M. et al. "Islanding the power grid on the transmission level: less connections for more security" *Sci Rep* 6, 34797 (2016). <https://doi.org/10.1038/srep34797>

Author Reply: Thank you for this comment. We have included the citation of this work and expanded our initial claim on inverse Braess' paradox:

"First results on inverse Braess' paradox stabilizing islanded grids have already been ob-

served [15].”

-
- [1] Shapiro, L. W. An electrical lemma. *Mathematics Magazine* **60**, 36–38 (1987). URL <https://doi.org/10.1080/0025570X.1987.11977274>.
- [2] Kaiser, F. & Witthaut, D. Topological theory of resilience and failure spreading in flow networks. *Phys. Rev. Research* **3**, 023161 (2021). URL <https://link.aps.org/doi/10.1103/PhysRevResearch.3.023161>.
- [3] Witthaut, D. & Timme, M. Braess’s Paradox in Oscillator Networks, Desynchronization and Power Outage. *New Journal of Physics* **14**, 083036 (2012).
- [4] Manik, D., Witthaut, D. & Timme, M. Predicting braess’ paradox in supply and transport networks. *arXiv preprint arXiv:2205.14685* (2022).
- [5] Maas, G., Bial, M. & Fijalkowski, J. Final report-system disturbance on 4 november 2006. *Union for the Coordination of Transmission of Electricity in Europe, Tech. Rep 1* (2007).
- [6] Cohen, J. E. & Horowitz, P. Paradoxical behaviour of mechanical and electrical networks. *Nature* **352**, 699 (1991).
- [7] Coletta, T. & Jacquod, P. Linear stability and the braess paradox in coupled-oscillator networks and electric power grids. *Physical Review E* **93**, 032222 (2016).
- [8] Nagurney, L. S. & Nagurney, A. Physical proof of the occurrence of the braess paradox in electrical circuits. *EPL (Europhysics Letters)* **115**, 28004 (2016).
- [9] Schultz, P., Heitzig, J. & Kurths, J. A random growth model for power grids and other spatially embedded infrastructure networks. *The European Physical Journal. Special Topics* **223**, 2593–2610 (2014).
- [10] Manik, D. Code to Reproduce Results on Braes Paradox in Electrical Power Grid Models (2022). URL <https://doi.org/10.5281/zenodo.6363078>.
- [11] Wood, A. J., Wollenberg, B. F. & Sheblé, G. B. *Power Generation, Operation and Control* (John Wiley & Sons, New York, 2013).
- [12] Rahmani, M., Kargarian, A. & Hug, G. Comprehensive power transfer distribution factor model for large-scale transmission expansion planning. *IET Generation, Transmission & Distribution* **10**, 2981–2989 (2016).
- [13] Ronellenfitsch, H., Timme, M. & Witthaut, D. A dual method for computing power transfer

- distribution factors. *IEEE Transactions on Power Systems* **32**, 1007–1015 (2016).
- [14] Shapiro, L. W. An electrical lemma. *Mathematics Magazine* **60**, 36–38 (1987).
- [15] Mureddu, M., Caldarelli, G., Damiano, A., Scala, A. & Meyer-Ortmanns, H. Islanding the power grid on the transmission level: less connections for more security. *Scientific Reports* **6**, 1–11 (2016).

REVIEWERS' COMMENTS

Reviewer #1 (Remarks to the Author):

Thanks for the responses. I have no more comments.

Reviewer #2 (Remarks to the Author):

I recommend the publication of the paper NCOMMS-21-37524A after a minor revision.

The authors refer to ref. 11 written in German, since they often use the paradox in that paper.

It is certainly helpful for the reader that there is also an English version of that paper:

On a paradox of traffic planning. (joint translation by Braess, D., Nagurney, A. and Wakolbinger, T.)
Transportation Science 39, 446-450 (2005) I recommend that this hint is added.

Reviewer #3 (Remarks to the Author):

I truly appreciate the diligence and effort of the authors in replying to all my several comments and questions. Even if the authors decided to relegate some of the technical results to a separate preprint, I believe the revised manuscript significantly improved with respect to the original one and I now recommend it for publication.

Reviewer #4 (Remarks to the Author):

The authors have satisfactorily answered to the comments

Response to the comments by the reviewers

We would like to thank all reviewers for their positive feedback and their recommendation to publish.

Response to Reviewer #2

Reviewer #2 (Remarks to the Author): *I recommend the publication of the paper NCOMMS-21-37524A after a minor revision.*

The authors refer to ref. 11 written in German, since they often use the paradox in that paper. It is certainly helpful for the reader that there is also an English version of that paper: On a paradox of traffic planning. (joint translation by Braess, D., Nagurney, A. and Wakolbinger, T.) Transportation Science 39, 446-450 (2005) I recommend that this hint is added.

Author Reply: We thank the reviewer for this suggestion and have cited the suggested paper in the revised article (reference number 12).

Response to Reviewers #1, #3 and #4

Reviewer #1 (Remarks to the Author):

Thanks for the responses. I have no more comments.

Reviewer #3 (Remarks to the Author):

I truly appreciate the diligence and effort of the authors in replying to all my several comments and questions. Even if the authors decided to relegate some of the technical results to a separate preprint, I believe the revised manuscript significantly improved with respect to the original one and I now recommend it for publication.

Reviewer #4 (Remarks to the Author):

The authors have satisfactorily answered to the comments

Author Reply: We thank the reviewers for their positive assessment.